# Hydrogen Sulfide (H_2_S) and Polysulfide (H_2_S_n_) Signaling: The First 25 Years

**DOI:** 10.3390/biom11060896

**Published:** 2021-06-16

**Authors:** Hideo Kimura

**Affiliations:** Faculty of Pharmaceutical Sciences, Sanyo-Onoda City University, Yamaguchi 756-0884, Japan; kimura@rs.socu.ac.jp

**Keywords:** hydrogen sulfide, polysulfides, *S*-sulfuration, nitric oxide, hydrogen peroxide, *S*-nitrosylation, *S*-sulfenylation, 3MST

## Abstract

Since the first description of hydrogen sulfide (H_2_S) as a toxic gas in 1713 by Bernardino Ramazzini, most studies on H_2_S have concentrated on its toxicity. In 1989, Warenycia et al. demonstrated the existence of endogenous H_2_S in the brain, suggesting that H_2_S may have physiological roles. In 1996, we demonstrated that hydrogen sulfide (H_2_S) is a potential signaling molecule, which can be produced by cystathionine β-synthase (CBS) to modify neurotransmission in the brain. Subsequently, we showed that H_2_S relaxes vascular smooth muscle in synergy with nitric oxide (NO) and that cystathionine γ-lyase (CSE) is another producing enzyme. This study also opened up a new research area of a crosstalk between H_2_S and NO. The cytoprotective effect, anti-inflammatory activity, energy formation, and oxygen sensing by H_2_S have been subsequently demonstrated. Two additional pathways for the production of H_2_S with 3-mercaptopyruvate sulfurtransferase (3MST) from l- and d-cysteine have been identified. We also discovered that hydrogen polysulfides (H_2_S_n_, n ≥ 2) are potential signaling molecules produced by 3MST. H_2_S_n_ regulate the activity of ion channels and enzymes, as well as even the growth of tumors. *S*-Sulfuration (*S*-sulfhydration) proposed by Snyder is the main mechanism for H_2_S/H_2_S_n_ underlying regulation of the activity of target proteins. This mini review focuses on the key findings on H_2_S/H_2_S_n_ signaling during the first 25 years.

## 1. Identification of H_2_S as a Signaling Molecule

Patients that recover from H_2_S poisoning show cognitive decline, and the levels of neurontransmitters in the brains of animals exposed to H_2_S change, suggesting that the brain is vulnerable to H_2_S toxicity [1]. Warenycia et al. measured the levels of H_2_S accumulated in the brain of rats exposed to H_2_S when they discovered a certain amount of H_2_S in the brain even without exposure to H_2_S [2]. Although the concentrations were overestimated, the existence of endogenous H_2_S was identified in the brain.

Pyridoxal 5′-phosphate-dependent enzymes, cystathionine β-synthase (CBS) and cystathionine γ-lyase (CSE), have been suggested to regulate several pathways. CBS catalyzes the first step of the transsulfuration pathway in which cystathionine is produced from serine and homocysteine, and cystathionine is further catalyzed by CSE to cysteine. An alternate pathway exists in which CBS catalyzes the condensation of cysteine with homocysteine to generate cystathionine and H_2_S [3,4]. CSE catalyzes an elimination reaction which metabolizes cysteine to pyruvate, NH_3_, and H_2_S [3,4]. However, rather than being recognized as a physiologically active molecule, in these early studies, H_2_S was merely thought to be a byproduct of the metabolic pathways.

The observations that H_2_S is produced by enzymes and exists in the brain prompted us to study a physiological role of this molecule. The activities of CBS and CSE have been intensively studied in the liver and kidney, but little is known about them in the brain. We found CBS in the brain and confirmed the production of H_2_S, which is augmented by *S*-adenosyl methionine (SAM) [5].

Other gaseous signaling molecules NO and carbon monoxide (CO) induce hippocampal long-term potentiation (LTP), a synaptic model of memory formation, as retrograde messengers, which are produced at postsynapse and released to presynapse to facilitate a release of a neurotransmitter glutamate from presynapse [6,7,8,9,10]. We examined whether or not H_2_S has a similar effect. H_2_S facilitated the induction of LTP by enhancing the activity of *N*-methyl-d-aspartate (NMDA) receptors but not as a retrograde messenger [5].

NMDA receptors are activated by a reducing substance dithiothreitol (DTT) through the reduction of a cysteine disulfide bond located at the hinge of the ligand-binding domain [11]. Because H_2_S is a reducing substance, it is likely to be a mechanism for facilitating the induction of LTP. However, H_2_S with one-tenth of the concentration of DTT exerted a greater effect than that of DTT [5]. This observation suggested that there is an additional mechanism for LTP induction by H_2_S. The prominent neuroscientist Solomon Snyder commented the following in *Science News*: “They have very impressive evidence that H_2_S is a potential neurotransmitter. It is an exciting paper that should stimulate a lot of people’s interest” [12].

The synaptic transmission is regulated not only by events at synapses such as a release of transmitters and the sensitivity of receptors but also by astrocytes, a type of glia, which surround synapses. Astrocytes release gliotransmitters to regulate synaptic activity. We found that H_2_S induces Ca^2+^ influx in astrocytes, which was greatly suppressed by La^3+^, Gd^3+^_,_ and ruthenium red, broad-spectrum inhibitors known for transient receptor potential (TRP) channels, suggesting that H_2_S activates TRP channels [13]. H_2_S was reported to activate TRPA1 channels in urinary bladder and in sensory neurons, but concentrations greater than 1 mM were required for inducing responses [14,15].

## 2. Identification of H_2_S_n_ as Signaling Molecules

During this study, we found that a batch of NaHS, i.e., the sodium salt of H_2_S, with yellowish color was much more potent than the colorless batch. We successfully reproduced a solution with a similar color by dissolving elemental sulfur into Na_2_S solution according to a report by Searcy and Lee [16]. The color came from H_2_S_n_, which induces Ca^2+^ influx in astrocytes much more potently than H_2_S [17,18,19]. H_2_S_n_ are natural inorganic polymeric sulfur–sulfur species or sulfane sulfur, which we later found to be produced by 3-mercaptopyruvate sulfurtransferase (3MST) from 3-mercaptopyruvate [20,21,22] and the partial oxidation of H_2_S [19], such as via the chemical interaction with NO [23,24]. H_2_S_2_ (2.6 µM) exists in the brain almost equivalent to the level of H_2_S (3 µM) [25]. Ca^2+^ influx induced in astrocytes by AITC, cinnamaldehyde, selective activators of TRPA1 channels, and Na_2_S_3_ was greatly suppressed by HC030031 and AP-18, selective inhibitors of TRPA1 channels. In astrocytes transfected with TRPA1-siRNA, Ca^2+^ influx was not efficiently induced by Na_2_S_3_ [19]. The EC_50_ value for H_2_S was 116 µM, while that for H_2_S_3_ was 91 nM, suggesting that H_2_S_n_ rather than H_2_S are ligands for TRPA1 channels [13,17,18,19]. The amino terminus of TRPA1 channels has 24 cysteine residues [26], and two cysteine residues Cys422 and Cys634 are sensitive to H_2_S_n_ [27].

*S*-Sulfuration (*S*-sulfuhydration) was proposed by Snyder and colleagues to regulate the activity of target proteins by H_2_S [28]. This proposal needs a minor revision to highlight H_2_S_n_ but not H_2_S *S*-sulfurate cysteine residues. In contrast, H_2_S *S*-sulfurates oxidized cysteine residues such as those *S*-nitrosylated and *S*-sulfenylated [29]. H_2_S_n_ *S*-sulfurate (*S*-sulfhydrate) two cysteine residues of TRPA1 channels to induce the conformational changes to activate the channels. As an alternative mechanism, one cysteine residue, which is *S*-sulfurated, reacts with the remaining cysteine residue to generate a cysteine disulfide bond. Although the conformation has not been examined in detail, the latter mechanism may induce conformational changes more efficiently than the former one.

Various target proteins of H_2_S_n_ have been identified such as a tumor suppressor phosphate and tensin homolog (PTEN), protein kinase G1α, and an enzyme responsible for glycolysis, glyceraldehyde 3-phosphate dehydrogenase (GAPDH) [28,30,31,32]. It has been reported that GAPDH is activated by H_2_S through *S*-sulfuration of the active site Cys150 [28], while it is suppressed by H_2_S_n_ through *S*-sulfuration of Cys156, which is not the active site [32]. Cys150 may be an oxidized residue when *S*-nitrosylated or *S*-sulfenylated, which can be *S*-sulfurated by H_2_S, while Cys156 must be a thiol, which is *S*-sulfurated by H_2_S_n_.

## 3. Synergy and Crosstalk between H_2_S and NO

H_2_S relaxes vascular smooth muscle in synergy with NO [33]. A similar result was also obtained in the ileum [34]. Whiteman et al. proposed that the chemical interaction of H_2_S with NO generate nitrosothiol, which releases NO in the presence of Cu^2+^ [35]. Filipovic et al. reported that H_2_S and NO produces nitroxyl (HNO) as a major product, as well as H_2_S_n_ [36,37], while Cortese-Krott et al. suggested that SSNO^−^ as a major product with H_2_S_n_ as a minor one [38]. We proposed that H_2_S_n_ are major products [23]. The effect of H_2_S_n_ and that of the products obtained from the mixture of Na_2_S and diethylamine NONOate, an NO donor, were eliminated when they were exposed to cyanide or DTT [23]. In contrast, HNO is resistant to cyanide, and SSNO^−^ is resistant to DTT. Based on these observations, H_2_S_n_ are potential chemical entities produced from H_2_S and NO [23,37,38]. Bogdandi et al. recently suggested that H_2_S_n_ transiently activate TRPA1 channels at the early phase of the production from H_2_S and NO, while the more stable product SSNO^−^ sustainably activates the channels [39].

## 4. Vascular Tone Regulation by H_2_S and H_2_S_n_

Since H_2_S relaxes vascular smooth muscle in synergy with NO [33] and activates ATP-dependent K^+^ (K_ATP_) channels [40], it has been suggested that H_2_S is a potential endothelial-derived hyperpolarizing factor (EDHF), which is a component of endothelial-derived relaxing factor (EDRF) [41]. However, previous studies showed that the hyperpolarization induced by EDHF is resistant to glibenclamide, a K_ATP_ channel blocker [42,43]. The relaxation of vascular smooth muscle in the mesenteric bed, which is mediated predominantly by EDHF, is rather abolished by apamine, a blocker of Ca^2+^-activated K^+^ channels [44].

H_2_S_n_ are potential EDHFs (Figure 1). H_2_S_n_ produced by 3MST together with cysteine aminotransferase (CAT), both of which are localized to the vascular endothelium [20,45,46], or H_2_S_n_ generated by the chemical interaction between H_2_S and NO produced by endothelial NO synthase (eNOS) can activate TRPA1 channels [19,23] localized to myoendothelial junctions. The channels induce Ca^2+^ influx, which activate Ca^2+^-activated K^+^ channels to hyperpolarize the endothelial cell plasma membrane. The change in membrane potential is conducted via myoendothelial gap junctions to hyperpolarize the vascular smooth muscle [47].

H_2_S has also been demonstrated to relax vascular smooth muscle via the protein kinase G pathway as an endogenous inhibitor of phosphodiesterase and increases the levels of both cyclic GMP and cyclic AMP [48,49], as well as by activating Kv7 potassium channels [50]. Kv7 channels are also involved in CBS-derived H_2_S induced human malignant hyperthermia syndrome triggered by volatile inhalation anesthetics in skeletal muscle [51].

## 5. Cytoprotective Effect of H_2_S, H_2_S_n_, and H_2_SO_3_

The impression of H_2_S as toxic gas led to its cytoprotective effect being overlooked [52]. Expecting that all cells would be killed by H_2_S, I applied NaHS to cells and incubated for overnight. On the contrary, cells were lively and survived from the toxin. H_2_S increases the production of glutathione (GSH), a major intracellular antioxidant, by enhancing the activity of cystine/glutamate antiporter, which incorporates cystine into cells, and of glutamate cysteine ligase (GCL), a rate-limiting enzyme for GSH production [52,53]. H_2_S also facilitates the translocation of GSH into mitochondria [53]. The protective activity of H_2_S is also exerted through the stabilization of membrane potential by enhancing the activity of K_ATP_ channels and cystic fibrosys transmembrane conductance regulator (CFTR) Cl^−^ channels [54]. Lefer and colleagues demonstrated that H_2_S protects the heart from ischemia/reperfusion injury by preserving mitochondrial function [55].

H_2_S_n_ *S*-sulfurate Keap1 and release Nrf2 from the Keap1/Nrf2 complex to the nucleus, where Nrf2 upregulates antioxidant genes including the GCL gene to increase the production of GSH [56]. H_2_S_n_ increase the levels of GSH and protect cells from oxidative stress to a greater level than H_2_S [57]. Sulfite (H_2_SO_3_), a metabolite of H_2_S and H_2_S_n_, protects neurons by increasing the production of GSH as efficiently as the parental molecules [57].

## 6. Signaling by H_2_S, H_2_S_n_ through *S*-Sulfuration and Bound Sulfane Sulfur

In addition to CBS and CSE, 3MST, along with CAT or DAO, was recognized to produce H_2_S from l- or d-cysteine, respectively [46,58,59]. Subsequently, 3MST was found to produce H_2_S_n_ and other *S*-sulfurated molecules such as cysteine persulfide, GSSH, and *S*-sulfurated cysteine residues [20,21,60]. Other enzymes such as sulfide-quinone oxidoreductase (SQR), haemoglobin, neuroglobin, catalase, super oxide dismutase (SOD), cysteine tRNA synthetase (CARS), and peroxidases have been identified to produce H_2_S_n_ and other *S*-sulfurated molecules [61,62,63,64,65,66,67,68,69].

In total, 10–20% of cysteine residues of proteins are *S*-sulfurated [28], also observed as a part of bound sulfane sulfur, which releases H_2_S under reducing conditions, including H_2_S_n_, cysteine persulfide, GSSH, and *S*-sulfurated cysteine residues [70,71,72,73]. In cells and tissues, 5–12% of total protein cysteine residues are oxidized, such as *S*-nitrosylated (P-CysSNO) and *S*-sulfenylated (P-CysSOH), and this can be increased to more than 40% under oxidative conditions [74] (Figure 2). The amount of bound sulfane sulfur and its associated species is distinct among tissues. For example, heart homogenates release H_2_S under reducing conditions much less than those from the liver and the brain, while heart homogenates absorb H_2_S as fast as liver homogenates [73]. P-CysSNO and P-CysSOH react with H_2_S to generate P-CysSSH, while they do not release H_2_S under reducing conditions. These observations suggest that the heart may contain P-CysSNO and P-CysSOH more abundantly than the liver and the brain.

Some cysteine residues are oxidized by H_2_O_2_ to generate *S*-nitrosylated cysteine residues, and some others are *S*-nitrosylated by NO. These oxidized cysteine residues are *S*-sulfurated by H_2_S rather than H_2_S_n_ (Figure 2). Cys150 and Cys156 of GAPDH may be in different oxidation states, as described previously [28,32]. Zivanovic et al. demonstrated that the activity of manganese superoxide dismutase is suppressed through *S*-sulfenylation by H_2_O_2_, while its activity is recovered by H_2_S, which *S*-sulfurates the *S*-sulfenylated cysteine residues [75]. The same group showed that epidermal growth factor (EGF) activates its receptor, in which the levels of *S*-sulfenylated cysteine residues are increased at the early phase, while those of *S*-sulfurated residues are increased at the late phase when the expression of H_2_S-producing enzymes is enhanced. H_2_S *S*-sulfurates those *S*-sulfenylated cysteine residues to regulate their activity (Figure 2).

Another role of *S*-sulfuration is that it enables proteins to recover their functions from over-oxidization. Sulfinic (P-CysSO_2_H) and sulfonic acids (P-CysSO_3_H) are not reduced back to P-CysSH by thioredoxin and deteriorate the protein function. In contrast, *S*-sulfurated proteins P-CysSSO_2_H and P-CysSSO_3_H can be reduced by thioredoxin to P-CysSH [75,76].

## 7. Diseases Caused by the Disturbance of H_2_S and H_2_S_n_

Both an excess and a deficiency of H_2_S and H_2_S_n_ have been suggested in the pathogenesis of schizophrenia. Thiol homeostasis is shifted to oxidized conditions, reflecting significantly less H_2_S and more disulfide bond formation in patients than normal individuals [77,78]. In contrast, we suggested that excess H_2_S and H_2_S_n_ are involved in the pathogenesis. Mice with high expression of 3MST impaired prepulse inhibition, an endophenotype for schizophrenia, and 3MST levels were positively correlated with symptom severity scores [79].

CBS and H_2_S may be involved in regulating proliferation and bioenergetics in breast cancer, ovarian cancer, and colorectal cancer [80,81,82], and high levels of CBS, CSE, and 3MST expression were observed in lung cancer [83].

Gliomas with the highest grades of malignancy contained greater levels of polysulfides than glioma-free brain regions [84], and H_2_S_n_ levels were greater in glioblastoma-bearing regions than glioblastoma-free control regions [85]. In contrast, it was reported that CBS is involved in suppressing glioma, whereby glioma with suppressed CBS had high levels of VEGF and HIF-2α and was deeply invaded with dense vascularization and aggressive growth [86].

Parkinson’s disease is a neurodegenerative disorder. Parkin, an E3 ubiquitin ligase responsible for the clearance of misfolded proteins, is suppressed in this disease. Specific cysteine residues of parkin are *S*-nitrosylated in patients, while they are *S*-sulfurated in the normal individuals [87]. H_2_S may be involved in *S*-sulfurating the *S*-nitrosylated cysteine residues of parkin.

Down’s syndrome (DS) is characterized by impaired brain growth and maturation that causes mental retardation and is associated with an Alzheimer’s type of dementia in elderly adults. DS involves a trisomy of chromosome 21 where CBS is encoded, and its mRNA level is 12 times greater in myeloblasts of DS children, while CBS protein levels in the brains of patients are approximately three times greater compared to the normal individuals [88,89]. Higher levels of thiosulfate, a metabolite of H_2_S, were detected in patients [90]. Mice with CBS overexpression showed DS-like neurocognitive deficits [91]. Fibroblasts prepared from DS patients showed profound suppression of mitochondrial electron transport, oxygen consumption, and ATP generation [92].

Ethylmalonyl encephalopathy is an autosomal recessive early-onset disorder, defective in cytochrome c oxidase in the brain and muscle. In this disease, ETHE1, a gene encoding sulfur dioxygenase, which metabolizes H_2_S in collaboration with SQR, is deficient, and H_2_S levels are increased to suppress cytochrome c oxidase [93]. High levels of H_2_S and persulfides also suppress acyl-protein thioesterase in the mouse model of this disease [94,95].

The brain is very sensitive to oxygen deprivation, i.e., hypoxia. During hypoxia, heme oxygenase-2 produces less carbon monoxide, which suppresses the activity of CBS, resulting in the overproduction of H_2_S that stimulates the carotid body to increase in respiratory rate, heart rate, and blood pressure [96]. On the other hand, the increased levels of H_2_S, in turn, suppress cytochrome c oxidase to cause hypoxic brain injury [97]. The levels are augmented in SQR-deficient animal models [97].

## 8. Perspective

Cysteine residues of the target proteins initially found, such as TRPA1 channels, PTEN, and protein kinase G1α, may be thiols whose sulfur has the oxidation state of −2, and that of H_2_S_n_ has the oxidation state of −1 or 0 and is able to *S*-sulfurate thiols to regulate the activity of the targets. On the basis of these observations, *S*-sulfurated molecules including H_2_S_n_ have been recognized as the chemical entities as signaling molecules rather than H_2_S, which is thought to be a mere precursor of H_2_S_n_ or a byproduct of other *S*-sulfurated molecules. However, depending on the redox conditions of cysteine residues of target proteins, both H_2_S and *S*-sulfurated molecules including H_2_S_n_ can *S*-sulfurate the targets. Considering that the endogenous levels of H_2_S (approximately 3 µM in the brain) and H_2_S_n_ (2.6 µM) are well balanced [25] and that the reaction between these signaling molecules and targets is fast [21], both H_2_S and H_2_S_2_ may react with their corresponding targets at a similar frequency. Identifying the redox conditions of target cysteine residues may help understand the functions of H_2_S or H_2_S_n_ as signaling molecules for specific targets.

High concentrations of H_2_S are toxic, while low concentrations are beneficial. Examples of the former can be observed in Down’s syndrome, ethylmalonyl encephalopathy, and hypoxic brain injury, while examples of the latter can be observed in Parkinson’s disease and Huntington’s disease [87,92,93,96,97,98]. Similarly, H_2_S exerts opposite effects depending on its concentration as observed in the effect of acetylcholine on the vascular smooth muscle, where low concentrations exert relaxation, while high concentrations exert contraction. Acetylcholine was previously known to contract vasculature, before Furchgott and Zawadzki discovered that lower concentrations of acetylcholine relaxed vasculature where endothelial cells were intact to release EDRF (NO) [99]. Recently, Vellecco et al. identified that vascular contraction by low concentrations of H_2_S (10 nM to 3 µM) is mediated by cyclic IMP [100]. The effect of nanomolar concentrations of H_2_S has also been found in T-cell activation [101]. The target, which responds to low concentrations of H_2_S, should be compared with its response to H_2_S_n_. It will be interesting to know whether or not the targets are oxidized cysteine residues.

Many compounds which release H_2_S and H_2_S_n_ have been developed for clinical use on the basis of their cytoprotective and anti-inflammatory effects. Some of them have successfully completed phase 2 clinical trials and are proceeding further, while many preclinical compounds are awaiting trials [102,103]. It is hoped that various H_2_S-based compounds will be translated into the field of clinical therapy over the next decade.

## Figures and Tables

**Figure 1 biomolecules-11-00896-f001:**
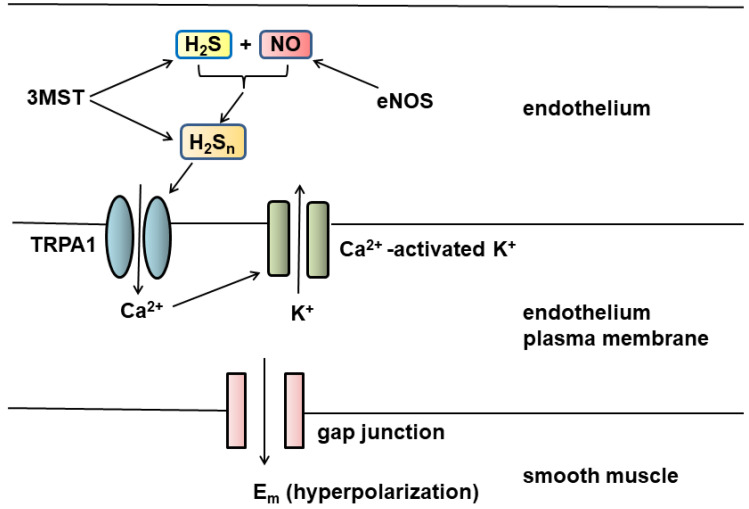
H_2_S_n_ are potential EDHFs. Both 3MST and eNOS are localized to endothelium. H_2_S_n_ produced by 3MST or by the chemical interaction between H_2_S and NO activate TRPA1 channels present in myoendothelial junctions to induce Ca^2+^ influx, which activates Ca^2+^-dependent K^+^ channels. The change in membrane potential is conducted via gap junction to hyperpolarize the smooth muscle plasma membrane.

**Figure 2 biomolecules-11-00896-f002:**
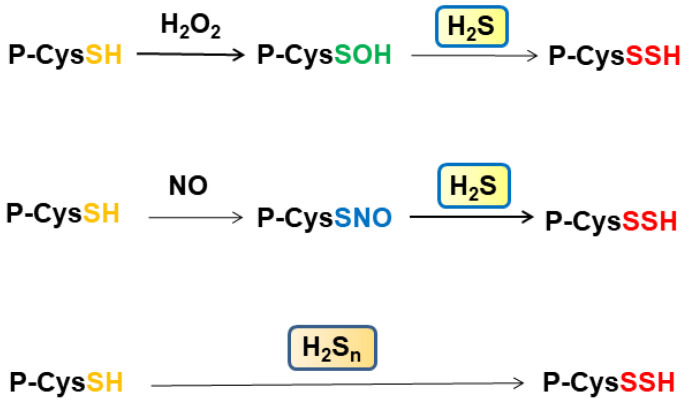
*S*-Sulfuration of cysteine residues by H_2_S and H_2_S_n_. Cysteine residues are *S*-sulfenylated by H_2_O_2_ and *S*-nitrosylated by NO. These oxidized cysteine residues are *S*-sulfurated by H_2_S. In contrast, cysteine residues are *S*-sulfurated by H_2_S_n_.

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
