# Peer review of "Hydrogen Sulfide (H2S) and Polysulfide (H2Sn) Signaling: The First 25 Years"

_biomolecules, 2021, doi:10.3390/biom11060896_

Round 1

Reviewer 1 Report

This review article is well written and designed and briefly describes the history of H2S from the discovery to the last insights.

I recommend its acceptation after the correction of some typos in the following sections:

  1. Abstract: the correct name of Ramazzini is Bernardino. Please, replace Bernerdino with Bernardino.
  2. Identification of H2S as a Signaling Molecule: Please replace Pyridoxal 5’-phosphate-depndent with “dependent”
  3. Synergy and cross talk between H2S and NO: The author mentioned some H2S-targets such as KATP channels but did not mentioned more recently discovered targets such as Kv7 channels and PDE, that are very important especially at vascular level. Probably the panorama could be more complete if also the more recent discovered targets will be described.
  4. Perspective: row 228: H2S needs to the “2” subscript (H2S)

Author Response

This review article is well written and designed and briefly describes the history of H2S from the discovery to the last insights.

I recommend its acceptation after the correction of some typos in the following sections:

  1. Abstract: the correct name of Ramazzini is Bernardino. Please, replace Bernerdino with Bernardino.
  2. Identification of H2S as a Signaling Molecule: Please replace Pyridoxal 5’-phosphate-depndent with “dependent”
  3. Synergy and cross talk between H2S and NO: The author mentioned some H2S-targets such as KATP channels but did not mentioned more recently discovered targets such as Kv7 channels and PDE, that are very important especially at vascular level. Probably the panorama could be more complete if also the more recent discovered targets will be described.
  4. Perspective: row 228: H2S needs to the “2” subscript (H2S)

Answer: Thank you for your comments. Kv7 and PDE in vascular smooth muscle and other recently discovered targets were discussed in the revised manuscript. Typos were corrected according to the reviewer’s suggestion.

H2S has also been demonstrated to relax vascular smooth muscle via the protein kinase G pathway as an endogenous inhibitor of phosphodiesterase and increases the levels of both cyclic GMP and cyclic AMP [48, 49] as well as by activating Kv7 potassium channels [50]. Kv7 channels are also involved in CBS-derived H2S induced human malignant hyperthermia syndrome triggered by volatile inhalation anesthetics in skeletal muscle [51].

The brain is very sensitive to oxygen deprivation, hypoxia. During hypoxia, heme oxygenase-2 produces less carbon monoxide, which suppresses the activity of CBS, resulting in the over production of H2S that stimulates the carotid body to increase in respiratory rate, heart rate and blood pressure [96]. On the other hand, the increased levels of H2S, in turn, suppress cytochrome c oxidase to cause hypoxic brain injury [97]. It is augmented in SQR deficient animal models [97].

Reviewer 2 Report

This is a well-written mini-review by the undeniable leader in this field describing how this area originated and has developed historically, much to the credit of the author.  There really isn’t anything of concern except maybe the quality of figure 1.  It didn’t appear to be lined up horizontally and the quality could be improved.

Author Response

This is a well-written mini-review by the undeniable leader in this field describing how this area originated and has developed historically, much to the credit of the author.  There really isn’t anything of concern except maybe the quality of figure 1.  It didn’t appear to be lined up horizontally and the quality could be improved.

Answer: Thank you for your comments. According to the reviewer’s suggestion the quality of Fig. 1 was improved.

Reviewer 3 Report

Hideo Kimura presented a review on Hydrogen sulfide (H2S) and polysulfide (H2Sn) signaling over the last 25 years.

The present review is of considerable interest but has one major drawback: a large part of the known literature has not been considered and the author mainly focused on his own research findings omitting a good part of other interesting papers. There is a recent good review on Molecules 2019 Apr 6;24(7):1359. doi: 10.3390/molecules24071359. which provides further reading and important concepts regarding H2Sn 

Minor points:

It is missing a general introduction to H2Sn 

The conclusions are less an expert opinion and outlook but rather additional fidings, thus this section should be improved

The figure 2 has no good underscores, thus Sn coud be missunderstood as tin instead of Sn

Author Response

Hideo Kimura presented a review on Hydrogen sulfide (H2S) and polysulfide (H2Sn) signaling over the last 25 years.

The present review is of considerable interest but has one major drawback: a large part of the known literature has not been considered and the author mainly focused on his own research findings omitting a good part of other interesting papers. There is a recent good review on Molecules 2019 Apr 6;24(7):1359. doi: 10.3390/molecules24071359. which provides further reading and important concepts regarding H2Sn 

Answer: Thank you for your comments. The suggested paper and other interesting reports were cited and discussed in the revised manuscript. Please also see the answer to Reviewer #1.

Minor points:

It is missing a general introduction to H2Sn 

Answer: A general introduction to H2Sn was added.

H2Sn are natural inorganic polymeric sulfur-sulfur species, or sulfane sulfur, which we later found to be produced by 3-mercaptopyruvate sulfurtransferase (3MST) from 3-mercaptopyruvate [20-22] and the partial oxidation of H2S [19] such as by the chemical interaction with NO [23, 24]. H2S2 (2.6 mM) exists in the brain almost equivalent to the level of H2S (3 mM) [25].

The conclusions are less an expert opinion and outlook but rather additional fidings, thus this section should be improved

Answer: The opinion and outlook were added and improved in the conclusions.

Cysteine residues of the target proteins initially found such as TRPA1 channels, PTEN and protein kinase G1a may be thiols whose sulfur has the oxidation state of -2, and that of H2Sn has oxidation state of -1 or 0 and is able to S-sulfurate thiols to regulate the activity of the targets. Based on these observations S-sulfurated molecules including H2Sn have been recognized as the chemical entities as signaling molecules rather than H2S, which is thought to be a mere precursor of H2Sn or byproduct of other S-sulfurated molecules. However, depending on the redox conditions of cysteine residues of target proteins, both H2S and S-sulfurated molecules including H2Sn can S-sulfurate the targets. Considering the endogenous levels of H2S (approximately 3 mM in the brain) and H2Sn (2.6 mM) are well balanced [25] and that the reaction between these signaling molecules and targets is fast [21], both H2S and H2S2 may react with their corresponding targets in a similar frequency. To identify the redox conditions of target cysteine residues may help understand either H2S or H2Sn functions as a signaling molecule for each specific target.

     High concentrations of H2S are toxic, while low concentrations are beneficial. Former examples are observed in Down’s syndrome, ethylmalonyl encephalopathy, and hypoxic brain injury, while those latter are in Parkinson’s disease, and Huntington’s disease [87, 92, 93, 96-98]. Similarly, H2S exerts the opposite effects depending on its concentrations as observed in the effect of acetylcholine on the vascular smooth muscle where low concentrations exert relaxation, while high concentrations do contraction. Acetylcholine had been known to contract vasculature, and then Furchgott and Zawadzki discovered that the lower concentrations of acetylcholine relaxed vasculature whose endothelial cells were intact to release EDRF (NO) [99]. Recently, Vellecco et al. identified the vascular contraction by low concentrations of H2S (10 nM to 3 mM) that is mediated by cyclic IMP [100]. The effect of nanomolar concentrations of H2S has also been found in T cell activation [101]. The target, which responds to low concentrations of H2S, should be compared with its response to H2Sn. It is interesting to know whether or not the targets are oxidized cysteine residues.

The figure 2 has no good underscores, thus Sn coud be missunderstood as tin instead of Sn

Answer: Fig. 2 was corrected according to the reviewer’s suggestion.

Reviewer 4 Report

Dr. Kimura is one of the pioneers of the hydrogen sulfide (H2S) field. The demonstration that H2S is a potential signaling molecule, which can be produced by cystathionine ß-synthase (CBS), to modify the neurotransmission in the brain was from the author’s laboratory.

This review highlights the significant findings of H2S and polysulfides as signaling molecules, it also hones on the cross-talk between H2S and nitric oxide, the first endogenous gasotransmitter.

Author Response

Dr. Kimura is one of the pioneers of the hydrogen sulfide (H2S) field. The demonstration that H2S is a potential signaling molecule, which can be produced by cystathionine ß-synthase (CBS), to modify the neurotransmission in the brain was from the author’s laboratory.

This review highlights the significant findings of H2S and polysulfides as signaling molecules, it also hones on the cross-talk between H2S and nitric oxide, the first endogenous gasotransmitter.

Answer: Thank you for your comments.

Round 2

Reviewer 3 Report

The authors improved the manuscript according to my suggestions, thus I suggest to accept it.